# On Comparing and Assessing Robustness of Some Popular Non-Stationary BINAR(1) Models

**Yuvraj Sunecher [1] and Naushad Mamode Khan [2,\*]**

1 Department of Accounting and Finance, University of Technology Mauritius, Pointe-Aux-Sables 11110, Mauritius; ysunecher@utm.ac.mu
2 Department of Economics and Statistics, University of Mauritius, Reduit 80835, Mauritius
\* Correspondence: n.mamodekhan@uom.ac.mu

**Abstract:** Intra-day transactions of stocks from competing firms in the financial markets are known to exhibit significant volatility and over-dispersion. This paper proposes some bivariate integer-valued auto-regressive models of order 1 (BINAR(1)) that are useful to analyze such financial series. These models were constructed under both time-variant and time-invariant conditions to capture features such as over-dispersion and non-stationarity in time series of counts. However, the quest for the most robust BINAR(1) models is still on. This paper considers specifically the family of BINAR(1)s with a non-diagonal cross-correlation structure and with unpaired innovation series. These assumptions relax the number of parameters to be estimated. Simulation experiments are performed to assess both the consistency of the estimators and the robust behavior of the BINAR(1)s under mis-specified innovation distribution specifications. The proposed BINAR(1)s are applied to analyze the intra-day transaction series of AstraZeneca and Ericsson. Diagnostic measures such as the root mean square errors (RMSEs) and Akaike information criteria (AICs) are also considered. The paper concludes that the BINAR(1)s with negative binomial and COM–Poisson innovations are among the most suitable models to analyze over-dispersed intra-day transaction series of stocks.

**Keywords:** BINAR(1); over-dispersion; CML; non-stationarity; diagnostics

## 1. Introduction

Initially, Pedeli and Karlis (2011) presented an extension of the classical integer-valued auto-regressive process of order 1 (INAR(1)) by assuming two inter-related simple INAR(1) processes with paired innovation series (BINAR(1)). Notably, in both the simple uni- and bivariate auto-regressive discrete-valued processes, the models consist of the survivor part which connects the current counting time series with its previous lagged observations and the respective error or innovation terms. In fact, in their proposed BINAR(1) model, the cross-correlation between the two series was induced solely by the paired innovation terms, yielding a diagonal BINAR structure. Based on this seminal paper, further developments in BINAR(1) models have occurred, particularly with different types of innovations Sajjadnia et al. (2023), Sharafi et al. (2023), and Pascual and Akhundjanov (2021)), with various thinning operators Ristić et al. (2009, 2012, and with different mechanisms to model the inter-relationship between the two series (refer to Yang et al. (2023) and Chen et al. (2022)). This paper lays emphasis on the alternative representations of the BINAR(1) process with different innovation terms and a non-diagonal BINAR structure for capturing the dependence between the series.

In fact, Pedeli and Karlis (2013a) extended the proposed BINAR(1) model in Pedeli and Karlis (2011) by adding survivor terms from the counter series and allowing paired innovation terms. By allowing for two sources of cross-correlation, the number of parameters to be estimated increases and complicates the estimation procedure. In addition, the marginal distribution of the counting series under such assumptions becomes difficult to identify.

On the other hand, Ristic et al. (2012) and Nastic et al. (2016) proposed a similar BINAR(1) as Pedeli and Karlis (2013a) but based on negative binomial thinning and independent innovation terms. This modeling approach relaxes some assumptions on the innovation distributions and reduces the number of parameters to be estimated. Nastic et al. (2016) proved that their proposed BINAR(1) provides better AICs than Pedeli and Karlis (2013a) and the simple BINAR(1) in Pedeli and Karlis (2011).

However, these models have mostly been developed under strict stationary or time-independent conditions. Mamode Khan et al. (2016) proposed a first BINAR(1) process with paired Poisson innovations with time-dependent marginal moments. This led the way to other such BINAR(1) processes with paired negative binomial (NB) and COM–Poisson (CMP) innovations Jowaheer and Sutradhar (2002); Mamode Khan et al. (2016); Shmueli et al. (2005); Sunecher et al. (2017). However, these BINAR(1)s consider only the diagonal cross-correlation structure. This paper, therefore, extends the work of Nastic et al. (2016) by considering the non-diagonal cross-correlation structure under time-variant moments. In addition, we compare BINAR(1) processes with different innovation specifications and identify the most robust BINAR process under mis-specified innovation distributions. The paper is organized as follows: In Section 2, we present the non-stationary BINAR(1) model with non-diagonal cross-correlation structure and its properties. Section 3 provides details on the inferential approach to obtain the model parameters in Section 2. The simulation study in Section 4 is split into two parts: under some time-dependent covariate design, some BINAR(1) series is generated with different innovation distributions. In the next subsection, data are generated under a specified BINAR(1) but are fitted using a BINAR process with a different innovation distribution. This subsection specifically measures the relative efficiencies between the two BINAR processes. In Section 5, the different BINAR(1)s are applied to analyze the AstraZeneca and Ericsson time series. The conclusions are presented in Section 6.

## 2. Construction of BINAR(1) Model

Consider the random variable $(Y_t^{[1]}, Y_t^{[2]})$, $t = 1, 2, 3, \ldots, T$, where

$$\begin{pmatrix} Y_t^{[1]} \\ Y_t^{[2]} \end{pmatrix} = \begin{pmatrix} \rho_{11} & \rho_{12} \\ \rho_{21} & \rho_{22} \end{pmatrix} \circ \begin{pmatrix} Y_{t-1}^{[1]} \\ Y_{t-1}^{[2]} \end{pmatrix} + \begin{pmatrix} R_t^{[1]} \\ R_t^{[2]} \end{pmatrix} \tag{1}$$

Here, $R_t^{[k]}$, $k \in \{1, 2\}$ is the random error term at the $t^{th}$ time point for the $k^{th}$ series, $\rho_{jk}$, for $j, k \in \{1, 2\}$, is constant over $\in [1, 2]$, and '$\circ$' denotes the binomial thinning operator such that

$$\rho_{jk} \circ Y_{t-1} = \begin{cases} \sum_{t=1}^{Y_{t-1}^{[k]}} b_l(\rho_{jk}), & Y_{t-1}^{[k]} > 0, \\ 0 & Y_{t-1}^{[k]} = 0 \end{cases} \tag{2}$$

where $b_l$ is a binary random variable, with $P(b_l = 1) = \rho_{jk} = 1 - P(b_l = 0)$.

The assumptions underlying the above model are:

(1) $R_t^{[k]}$ is assumed to be independent and identically distributed with mean $\lambda_t^{[k]}$ and variance $\nu_k \lambda_t^{[k]}$ with $\nu_k > 0$, such that for $\nu = 1$, $R_t$ is Poisson $(\lambda_t^{[k]})$ and for $\nu_k > (<)1$, $R_t$ is over (under)-dispersed. Hence, in the above setup, we present a general distributional form of $R_t^{[k]}$ such that for some specific representation of $E(R_t^{[k]})$ and $Var(R_t^{[k]})$, $R_t^{[k]}$ can be shown to follow some popular discrete distributions. For the special case of over-dispersion, this is discussed in Section 4.

(2)

$$Cov(Y_t^{[k]}, R_t^{[k]}) = \begin{cases} Var(R_t^{[k]}), & t = u, \\ 0, & t \neq u, \end{cases}$$

for $\{t, u\} = 1, 2, 3, \ldots, T$, and hence, $Cov(Y_t^{[1]}, R_{t+h}^{[2]}) = 0$ for any $h \in \mathbb{Z}^+$.

(3)  The pair $(R_t^{[1]}, R_t^{[2]})$ is independent, and hence, the inter-relation between $\mathrm{Cov}(Y_t^{[1]}, Y_t^{[2]})$ is induced by the previous-lagged terms, and hence,

$$\sigma_{t,1,2} = \rho_{11}\rho_{21}\sigma_{t-1,1,1} + \rho_{12}\rho_{22}\sigma_{t-1,2,2} + \rho_{11}\rho_{22}\sigma_{t-1,1,2} + \rho_{12}\rho_{21}\sigma_{t-1,2,1}, \qquad (3)$$

where $\sigma_{t,j,k}$ represents the covariance between $Y_t^{[j]}$ and $Y_t^{[k]}$ for $\{j,k\} \in \{1,2\}$.

Under these conditions, it is worthwhile to derive some moments of the proposed BINAR(1) model using the properties of the binomial thinning properties from Steutel and Van Harn (1986) as

$$E(Y_t^{[1]}) = \mu_t^{[1]} = \rho_{11}\mu_{t-1}^{[1]} + \rho_{12}\mu_{t-1}^{[2]} + \lambda_t^{[1]}, \qquad (4)$$

$$\sigma_{t,1,1} = \mu_t^{[1]} + \rho_{11}^2(\sigma_{t-1,1,1} - \mu_{t-1}^{[1]}) + \rho_{12}^2(\sigma_{t-1,2,2} - \mu_{t-1}^{[2]}) + (\nu_1 - 1)\lambda_t^{[1]}, \qquad (5)$$

and, similarly we have

$$E(Y_t^{[2]}) = \mu_t^{[2]} = \rho_{21}\mu_{t-1}^{[1]} + \rho_{22}\mu_{t-1}^{[2]} + \lambda_t^{[2]}, \qquad (6)$$

$$\sigma_{t,2,2} = \mu_t^{[2]} + \rho_{21}^2(\sigma_{t-1,1,1} - \mu_{t-1}^{[1]}) + \rho_{22}^2(\sigma_{t-1,2,2} - \mu_{t-1}^{[2]}) + (\nu_2 - 1)\lambda_t^{[2]}, \qquad (7)$$

**Remark 1.** *(1)   It is clear from the above expressions that for $\nu_k \geq 1$ and for $t = 1, \ldots (T-1)$, if $\sigma_{t-1,k,k} > \mu_{t-1}^{[k]}$, then $Y_t^{[k]}$ is over-dispersed.*

*(2)   The marginal distribution of $Y_t^{[k]}$ is difficult to identify and is considered unknown, even for $\nu_1, \nu_2 = 1$ (Poisson innovations), but the conditional probability distribution $f(Y_t^{[k]}|Y_{t-1}^{[k]})$ can be derived using the binomial thinning and the convolution properties. Note that the distribution of the innovation terms needs to be known in order to specify the conditional maximum likelihood (CML) function. In the event that the distribution of the random innovation term is unknown, the CML expression is impossible and a quasi-likelihood estimation procedure is practically infeasible since the estimation of the dispersion parameter $\nu_k$ requires some complicated score.*

## 3. Estimation of Parameters

Further to the discussion in Section 2, this section proposes the CML approach to estimate the unknown parameters of model (1). Using Equation (1) and assuming $\lambda_t^{[k]} = \exp(x_t\beta^{[k]})$, where $\beta^{[k]}$ is a $(p \times 1)$ regression vector for the $k^{th}$ series, which implies the vector of unknown parameters is $\epsilon = (\rho_{11}, \rho_{12}, \rho_{21}, \rho_{22}, \nu_1, \nu_2)$ is a $2(p+3)$ vector, then using the convolution property, the conditional distribution function is given by

$$f(y_t^{[1]} y_t^{[2]}|y_{t-1}^{[1]}, y_{t-1}^{[2]}) = \sum_{k=0}^{u} \sum_{j=0}^{v} f_1(k) f_2(s) P(R_t^{[1]} = y_t^{[1]} - k) P(R_t^{[2]} = y_t^{[2]} - s)) \qquad (8)$$

where $u = min(Y_t^{[1]} Y_{t-1}^{[1]})$ and $v = min(Y_t^{[2]} Y_{t-1}^{[2]})$. Here, we define

$$f_1(k) = \sum_{j_1=0}^{k} \binom{y_{t-1}^{[1]}}{j_1} \binom{y_{t-1}^{[2]}}{k-j_1} \rho_{11}^{j_1}(1-\rho_{11})^{y_{t-1}^{[1]}-j_1} \rho_{12}^{k-j_1}(1-\rho_{12})^{y_{t-1}^{[2]}-k+j_1} \qquad (9)$$

$$f_2(s) = \sum_{j_2=0}^{s} \binom{y_{t-1}^{[2]}}{j_2} \binom{y_{t-1}^{[1]}}{s-j_2} \rho_{22}^{j_2}(1-\rho_{22})^{y_{t-1}^{[2]}-j_2} \rho_{21}^{s-j_2}(1-\rho_{21})^{y_{t-1}^{[1]}-s+j_2} \qquad (10)$$

Then, we can write the conditional likelihood function as $L(\epsilon) = \prod_{t=2}^{T} f(Y_t^{[1]}, Y_t^{[2]}|y_{t-1}^{[1]}, Y_{t-1}^{[2]})$. The maximization of $L(\epsilon)$ is achieved using the standard *optim* routine in R with quasi-Newton approaches (BFGS). The asymptotic properties of the CML estimators are

established in Pedeli and Karlis (2011) and Franke and Rao (1995). The one-step-ahead expected mean of $Y_t^{[k]}|Y_{t-1}^{[k]}$ is given by

$$E(Y_{t+1}^{[1]}|y_t^{[1]}, y_t^{[2]}) = \hat{\rho}_{11}y_t^{[1]} + \hat{\rho}_{12}y_t^{[2]} + \hat{\lambda}_{t+1}^{[1]} \tag{11}$$

$$E(Y_{t+1}^{[2]}|y_t^{[1]}, y_t^{[2]}) = \hat{\rho}_{21}y_t^{[1]} + \hat{\rho}_{22}y_t^{[2]} + \hat{\lambda}_{t+1}^{[2]} \tag{12}$$

Here, we have $\hat{\lambda}_t^{[k]} = \exp(x_t\hat{\beta}^{[k]})$.

## 4. Simulation Study

### 4.1. Data Generating Processes and Results

This section presents simulation experiments where series of counts are generated using the BINAR(1) in Equation (1). We consider different distributions for the innovations $\{R_t^{[1]}, R_t^{[2]}\}$, specifically,

(1) BINAR(1)NB (negative binomial): The error terms $R_t^{[k]}$ are assumed to follow the ecological definition of the NB as illustrated in Jowaheer et al. (2017) such that $E(R_t^{[k]}) = \lambda_t^{[k]}$, $Var(R_t^{[k]}) = \lambda_t^{[k]} + c_k[\lambda_t^{[k]}]^2$, $c_k > 0$, and here, the index of over-dispersion is denoted by $c_k$.

(2) BINAR(1) CMP (COM–Poisson): Following Shmueli et al. (2005), under the COM–Poisson assumption, $E(R_t^{[k]}) = \lambda_t^{[k]} = \theta_t^{[k]1/\nu_k} - \frac{\nu_k-1}{2\nu_k}$ and $Var(R_t^{[k]}) = \frac{\theta_t^{[k]1/\nu_k}}{\nu_k}$. For $\nu_k < 1$, Shmueli et al. (2005) proved that $R_t^{[k]}$ is over-dispersed.

(3) BINAR(1)G (geometric): Using the geometric probability definition in Popovic et al. (2016), $E(R_t^{[k]}) = \lambda_t^{[k]}$ and $Var(R_t^{[k]}) = \lambda_t^{[k]} + [\lambda_t^{[k]}]^2$.

(4) For the BINAR(1)P (Poisson) model, refer to Section 2.

Note: The link functions in BINAR(1)NB, BINAR(1)G, and BINAR(1)P are commonly given by $\lambda_t^{[k]} = \exp(x_t\beta^{[k]})$, while for BINAR(1)CMP, $\theta_t^{[k]} = \exp(x_t\beta^{[k]})$. Under this connotation, we denote the link function by $\gamma_t^{[k]} = \exp(x_t\beta^{[k]})$. Suppose the covariate design is given by

$$x_{t1} = \begin{cases} 1 & (t = 1, \dots, T/4), \\ 2t & (t = (T/4)+1, \dots, 3T/4), \\ \cos(\frac{2\pi t}{6}) & (t = (3T/4)+1, \dots, T), \end{cases}$$

$$x_{t2} = \begin{cases} \sin(\frac{3\pi t}{12}) & (t = 1, \dots, T/4), \\ \cos(\frac{\pi t}{6}) & (t = (T/4)+1, \dots, 3T/4), \\ \sin(\frac{2\pi t}{6}) & (t = (3T/4)+1, \dots, T), \end{cases}$$

where $\gamma_t^{[k]} = \exp(\beta_1^{[k]}x_{t1} + \beta_2^{[k]}x_{t2})$, with $\beta_1^{[1]} = 1.5$ and $\beta_2^{[1]} = 1.00$, $\beta_1^{[2]} = -0.95$ and $\beta_2^{[2]} = 2.50$, $k \in \{1, 2\}$. $R_t^{[k]}$ is then generated with parameters $\lambda_t^{[k]}$. Note that for $t = 1$, $Y_t^{[k]} = R_t^{[k]}$, for $k \in \{1, 2\}$. Standard routines are used in R for simulating the Poisson and NB innovations. As for the COM–Poisson innovations $R_t^{[k]}$, the COM–Poisson package in R is used. For the above simulation study, we assume $c_1 = 0.5$; $c_2 = 0.8$ and $\nu_1 = 0.2$; $\nu_2 = 0.3$. The IOD represents the corresponding over-dispersed coefficient.

The results in Tables 1 and 2 based on 2000 replications for each combination present the mean estimate and the corresponding standard errors. If the values of the parameter estimators are examined for each model, they become closer to true values as T increases.

**Table 1.** Simulated mean estimates of the model parameters and the corresponding standard error (s.e.).

| $T$ | Model | $\hat{\beta}_1^{[1]}$ | $\hat{\beta}_2^{[1]}$ | $\hat{\beta}_1^{[2]}$ | $\hat{\beta}_2^{[2]}$ |
|---|---|---|---|---|---|
| 60 | BINAR(1)P | 1.4651 | 1.0233 | −0.9555 | 2.4521 |
| | | (0.5871) | (0.3881) | (0.4662) | (0.4012) |
| | BINAR(1)NB | 1.5221 | 1.0122 | −0.9488 | 2.5221 |
| | | (0.3892) | (0.3233) | (0.2671) | (0.3503) |
| | BINAR(1)G | 1.4771 | 0.9452 | −0.8956 | 2.6125 |
| | | (0.3561) | (0.2981) | (0.2892) | (0.3311) |
| | BINAR(1)CMP | 1.5501 | 1.1121 | −0.9231 | 2.415 |
| | | (0.3701) | (0.2302) | (0.2999) | (0.3560) |
| 200 | BINAR(1)P | 1.5230 | 0.9899 | −0.9476 | 2.5101 |
| | | (0.3775) | (0.2892) | (0.3579) | (0.3551) |
| | BINAR(1)NB | 1.4818 | 1.1210 | −0.9559 | 2.4818 |
| | | (0.3542) | (0.3045) | (0.2222) | (0.3198) |
| | BINAR(1)G | 1.5212 | 1.2320 | −0.9444 | 2.5321 |
| | | (0.2807) | (0.2616) | (0.2375) | (0.3042) |
| | BINAR(1)CMP | 1.4857 | 0.9875 | −0.9568 | 2.5320 |
| | | (0.2881) | (0.1974) | (0.2065) | (0.2762) |
| 600 | BINAR(1)P | 1.5101 | 1.0142 | −0.9555 | 2.4998 |
| | | (0.1542) | (0.1101) | (0.1457) | (0.2015) |
| | BINAR(1)NB | 1.4986 | 0.9972 | −0.9469 | 2.521 |
| | | (0.1066) | (0.0873) | (0.1066) | (0.1701) |
| | BINAR(1)G | 1.5091 | 0.9987 | −0.9561 | 2.5011 |
| | | (0.1226) | (0.2032) | (0.1653) | (0.1876) |
| | BINAR(1)CMP | 1.4988 | 1.0101 | −0.9499 | 2.4980 |
| | | (0.1501) | (0.1320) | (0.1503) | (0.1712) |

**Table 2.** Simulated mean estimates of the model parameters and the corresponding standard error (s.e).

| $T$ | Model | $\hat{\rho}_{11} = 0.9$ | $\hat{\rho}_{22} = 0.9$ | $\hat{\rho}_{12} = 0.3$ | $\hat{\rho}_{21} = 0.3$ | $IOD_1$ | $IOD_2$ |
|---|---|---|---|---|---|---|---|
| 60 | BINAR(1)P | 0.8890 | 0.9201 | 0.2881 | 0.3122 | | |
| | | (0.4522) | (0.3601) | (0.2832) | (0.2908) | | |
| | BINAR(1)NB | 0.9322 | 0.8899 | 0.3221 | 0.2987 | 0.5661 | 0.7871 |
| | | (0.2301) | (0.3281) | (0.2344) | (0.1981) | (0.3531) | (0.2809) |
| | BINAR(1)G | 0.8789 | 0.9102 | 0.2897 | 0.3159 | | |
| | | (0.2452) | (0.3566) | (0.2011) | (0.1892) | | |
| | BINAR(1)CMP | 0.9157 | 0.9162 | 0.2817 | 0.3139 | 0.2335 | 0.3187 |
| | | (0.2201) | (0.2872) | (0.1981) | (0.1976) | (0.2861) | (0.2472) |
| 200 | BINAR(1)P | 0.9101 | 0.8991 | 0.3201 | 0.2952 | | |
| | | (0.3709) | (0.2986) | (0.2329) | (0.2221) | | |
| | BINAR(1)NB | 0.8775 | 0.9212 | 0.2891 | 0.3312 | 0.4807 | 0.8122 |
| | | (0.1975) | (0.2605) | (0.1965) | (0.1652) | (0.2881) | (0.2227) |
| | BINAR(1)G | 0.9122 | 0.8829 | 0.3329 | 0.2899 | | |
| | | (0.1976) | (0.3017) | (0.1875) | (0.1563) | | |
| | BINAR(1)CMP | 0.8988 | 0.9035 | 0.3202 | 0.2980 | 0.2512 | 0.3331 |
| | | (0.1836) | (0.2356) | (0.1775) | (0.1652) | (0.2320) | (0.1854) |

**Table 2.** *Cont.*

| T | Model | $\hat{\rho}_{11} = 0.9$ | $\hat{\rho}_{22} = 0.9$ | $\hat{\rho}_{12} = 0.3$ | $\hat{\rho}_{21} = 0.3$ | $IOD_1$ | $IOD_2$ |
|---|---|---|---|---|---|---|---|
| 600 | BINAR(1)P | 0.8988 | 0.9042 | 0.2987 | 0.3015 | | |
| | | (0.1892) | (0.2042) | (0.1568) | (0.1892) | | |
| | BINAR(1)NB | 0.9015 | 0.8972 | 0.3052 | 0.2976 | 0.5023 | 0.7986 |
| | | (0.1237) | (0.2222) | (0.1380) | (0.1112) | (0.1019) | (0.1370) |
| | BINAR(1)G | 0.8972 | 0.9016 | 0.3110 | 0.2976 | | |
| | | (0.1131) | (0.1222) | (0.1065) | (0.1001) | | |
| | BINAR(1)CMP | 0.9015 | 0.8995 | 0.2989 | 0.3015 | 0.2508 | 0.3331 |
| | | (0.1001) | (0.1272) | (0.1329) | (0.1104) | (0.1076) | (0.1111) |

*4.2. Comparison Results*

For a given bivariate series of over-dispersed data, it is understood that BINAR(1)P, BINAR(1)NB, BINAR(1)G, and BINAR(1)CMP could be used for analysis, and thus, it is worth investigating the bias effect on the estimators of a BINAR(1) when the actual data are simulated by another of the above mentioned BINAR(1) processes. We start with BINAR(1)P simulated data and apply the other three BINAR(1) models. The relative efficiency (RE) ratios of the regression and thinning effects are reported, i.e., $RE_{W|M} = \frac{(s.e)BINAR(1)W}{(s.e)BINAR(1)M}$, where W is the working BINAR(1) process and M is the exact BINAR(1) process. The data are simulated using the same parameterization as above, with the same number of replications. The simulated mean REs assuming BINAR(1)P, BINAR(1)NB, BINAR(1)G, and BINAR(1)CMP models, are, respectively, given in Tables 3–6.

**Table 3.** Simulated mean REs assuming BINAR(1)P model.

| T | Model | $\hat{\beta}_1^{[1]}$ | $\hat{\beta}_2^{[1]}$ | $\hat{\beta}_1^{[2]}$ | $\hat{\beta}_2^{[2]}$ | $\hat{\rho}_{11} = 0.9$ | $\hat{\rho}_{22} = 0.9$ | $\hat{\rho}_{12} = 0.3$ | $\hat{\rho}_{21} = 0.3$ |
|---|---|---|---|---|---|---|---|---|---|
| 60 | $RE_{NB|P}$ | 0.5781 | 0.6777 | 0.5055 | 0.5981 | 0.6022 | 0.4609 | 0.6606 | 0.5909 |
| | $RE_{G|P}$ | 0.7880 | 0.8082 | 0.6522 | 0.7001 | 0.7881 | 0.6551 | 0.8090 | 0.7821 |
| | $RE_{CMP|P}$ | 0.6070 | 0.6602 | 0.6771 | 0.6231 | 0.6111 | 0.5662 | 0.6382 | 0.6444 |
| 200 | $RE_{NB|P}$ | 0.4301 | 0.3112 | 0.4211 | 0.5010 | 0.3651 | 0.4301 | 0.5001 | 0.3676 |
| | $RE_{G|P}$ | 0.5222 | 0.3201 | 0.4301 | 0.5661 | 0.4000 | 0.5212 | 0.5161 | 0.4331 |
| | $RE_{CMP|P}$ | 0.3667 | 0.4005 | 0.3991 | 0.4592 | 0.3881 | 0.3976 | 0.4892 | 0.4231 |
| 600 | $RE_{NB|P}$ | 0.2333 | 0.1143 | 0.3301 | 0.2209 | 0.1667 | 0.2998 | 0.3207 | 0.2337 |
| | $RE_{G|P}$ | 0.2706 | 0.2112 | 0.3015 | 0.3221 | 0.2561 | 0.3541 | 0.3441 | 0.2201 |
| | $RE_{CMP|P}$ | 0.1991 | 0.1200 | 0.3229 | 0.2159 | 0.1886 | 0.2341 | 0.2781 | 0.1980 |

As seen in Table 3, under BINAR(1)P simulated data, where the innovations are generated via Poisson model, the BINAR(1)NB and BINAR(1)CMP models seem to provide more efficient estimators. For larger time points, both BINAR(1)NB and BINAR(1)CMP model yield estimators with much lower standard errors than the BINAR(1)P model.

**Table 4.** Simulated mean REs assuming BINAR(1)NB model.

| $T$ | Model | $\hat{\beta}_1^{[1]}$ | $\hat{\beta}_2^{[1]}$ | $\hat{\beta}_1^{[2]}$ | $\hat{\beta}_2^{[2]}$ | $\hat{\rho}_{11}=0.9$ | $\hat{\rho}_{22}=0.9$ | $\hat{\rho}_{12}=0.3$ | $\hat{\rho}_{21}=0.3$ |
|---|---|---|---|---|---|---|---|---|---|
| 60 | $RE_{NB|P}$ | 1.2331 | 0.9890 | 1.6432 | 1.8995 | 0.8788 | 1.6678 | 1.3501 | 1.0111 |
| | $RE_{G|P}$ | 1.0101 | 0.9995 | 1.1105 | 0.9063 | 1.0001 | 1.0542 | 1.2110 | 1.0722 |
| | $RE_{CMP|P}$ | 0.8093 | 1.1551 | 0.9899 | 1.0001 | 0.9099 | 1.0651 | 0.9994 | 1.0311 |
| 200 | $RE_{NB|P}$ | 2.0541 | 1.6660 | 1.8988 | 1.4321 | 0.9899 | 1.2201 | 1.0871 | 1.5611 |
| | $RE_{G|P}$ | 1.0056 | 1.1221 | 1.0005 | 1.0043 | 0.9997 | 1.2022 | 1.0762 | 1.1167 |
| | $RE_{CMP|P}$ | 0.9996 | 1.0056 | 1.0121 | 0.9996 | 1.0233 | 1.0064 | 0.9988 | 1.0112 |
| 600 | $RE_{NB|P}$ | 1.3333 | 1.5042 | 1.2331 | 1.4320 | 1.1665 | 1.3990 | 0.9995 | 1.1335 |
| | $RE_{G|P}$ | 1.0441 | 1.0084 | 1.0771 | 0.9997 | 1.1782 | 1.0301 | 1.0014 | 1.0656 |
| | $RE_{CMP|P}$ | 0.9999 | 1.0002 | 0.9996 | 0.9098 | 0.9936 | 1.0352 | 0.9890 | 1.0012 |

It is clear that for the BINAR(1)NB model simulated series, the BINAR(1)P and BINAR(1)G model estimators are not as efficient as the BINAR(1)CMP, while the BINAR(1)CMP estimators have almost the same level of efficiency, in particular for a larger number of time points.

**Table 5.** Simulated mean REs assuming BINAR(1)G model.

| $T$ | Model | $\hat{\beta}_1^{[1]}$ | $\hat{\beta}_2^{[1]}$ | $\hat{\beta}_1^{[2]}$ | $\hat{\beta}_2^{[2]}$ | $\hat{\rho}_{11}=0.9$ | $\hat{\rho}_{22}=0.9$ | $\hat{\rho}_{12}=0.3$ | $\hat{\rho}_{21}=0.3$ |
|---|---|---|---|---|---|---|---|---|---|
| 60 | $RE_{NB|P}$ | 1.8991 | 1.0341 | 1.1167 | 1.8899 | 1.0952 | 1.1209 | 1.6667 | 1.2331 |
| | $RE_{G|P}$ | 0.8993 | 0.9088 | 0.9976 | 1.0012 | 0.9989 | 1.0022 | 0.9998 | 0.9552 |
| | $RE_{CMP|P}$ | 0.9191 | 1.0035 | 0.9899 | 1.0222 | 0.9997 | 0.9987 | 0.9890 | 1.0044 |
| 200 | $RE_{NB|P}$ | 1.5662 | 1.1120 | 1.0771 | 1.1242 | 1.1402 | 1.3333 | 1.3021 | 1.1101 |
| | $RE_{G|P}$ | 0.6707 | 0.4313 | 0.5662 | 0.8987 | 0.9562 | 0.9877 | 0.8917 | 0.9001 |
| | $RE_{CMP|P}$ | 0.7761 | 0.5612 | 0.6672 | 0.9631 | 0.9870 | 0.9660 | 0.8230 | 0.8652 |
| 600 | $RE_{NB|P}$ | 1.3241 | 1.1111 | 1.4321 | 1.2012 | 1.1555 | 1.4452 | 1.3201 | 1.0998 |
| | $RE_{G|P}$ | 0.4320 | 0.3309 | 0.3678 | 0.5555 | 0.7812 | 0.6702 | 0.6712 | 0.8688 |
| | $RE_{CMP|P}$ | 0.5420 | 0.3812 | 0.4552 | 0.5402 | 0.6612 | 0.7889 | 0.6809 | 0.7677 |

**Table 6.** Simulated mean REs assuming BINAR(1)CMP model.

| $T$ | Model | $\hat{\beta}_1^{[1]}$ | $\hat{\beta}_2^{[1]}$ | $\hat{\beta}_1^{[2]}$ | $\hat{\beta}_2^{[2]}$ | $\hat{\rho}_{11}=0.9$ | $\hat{\rho}_{22}=0.9$ | $\hat{\rho}_{12}=0.3$ | $\hat{\rho}_{21}=0.3$ |
|---|---|---|---|---|---|---|---|---|---|
| 60 | $RE_{NB|P}$ | 1.6998 | 1.8972 | 1.5132 | 1.6082 | 1.1928 | 1.6699 | 2.6781 | 1.4991 |
| | $RE_{G|P}$ | 1.0881 | 1.4331 | 1.3031 | 1.5421 | 1.1802 | 1.2115 | 1.5231 | 1.8802 |
| | $RE_{CMP|P}$ | 1.0561 | 1.2301 | 1.2778 | 1.4231 | 1.0112 | 1.2310 | 1.3112 | 1.2402 |
| 200 | $RE_{NB|P}$ | 1.2303 | 1.2102 | 1.2561 | 1.0321 | 1.0112 | 1.2221 | 1.2310 | 1.3683 |
| | $RE_{G|P}$ | 1.1011 | 1.3222 | 1.2314 | 1.0112 | 1.0231 | 1.0123 | 1.2301 | 1.3402 |
| | $RE_{CMP|P}$ | 1.0331 | 1.1111 | 1.2042 | 1.2356 | 1.1011 | 1.0042 | 1.2212 | 1.2212 |
| 600 | $RE_{NB|P}$ | 1.1333 | 1.0143 | 1.0332 | 1.0220 | 1.1321 | 1.0321 | 1.0221 | 1.0317 |
| | $RE_{G|P}$ | 1.0012 | 1.0102 | 1.0056 | 0.9989 | 1.0562 | 1.0510 | 1.0414 | 1.0201 |
| | $RE_{CMP|P}$ | 0.9995 | 1.0901 | 1.0209 | 0.9900 | 0.9967 | 1.0340 | 1.0781 | 1.0801 |

From Tables 5 and 6, it is clearly seen that under the true model BINAR(1)G, the BINAR(1)NB and BINAR(1)CMP estimators show some slight efficiency and robustness, while in Table 5, in some cases of T = 600, the BINAR(1)NB estimators almost share the same level of efficiency as BINAR(1)CMP. Thus, we could conclude from these tables that BINAR(1)NB and BINAR(1)CMP are appropriate to model over-dispersed bivariate series where the innovation distributions may be unknown.

## 5. Application to AstraZeneca and Ericsson Data Sets

In this section, we analyze the tick by tick AstraZeneca and Ericsson data using the different BINAR(1) models: BINAR(1)P, BINAR(1)NB, BINAR(1)G, and BINAR(1)CMP. These data are downloaded from the Ecovision system and represent the most frequently traded stocks on the Stockholmsbrsen, that opens at 09.30 and closes at 17.20. The AstraZeneca stock series data are collected for the period 2–22 July 2002, wherein the data in this study were aggregated into one-minute intervals of time and are based on minute 5501 to 5800, thus making 300 observations Quoreshi (2006, 2008).

As seen from examination of the data sets, some key descriptive statistics, including the mean, standard deviation, skewness, and kurtosis, in Table 7 are highly skewed. The minimum values of both the AstraZeneca and Ericsson series were recorded at 0.00 and the maximum values were at 17.00 and 64.00, respectively. During the sample period, the highest standard deviation value was recorded in the Ericsson and the lowest average value was in the AstraZeneca. Moreover, the plot of the Ericsson data is presented in Figure 1. Then, the autocorrelation function (ACF) and the partial ACF are, respectively, shown in Figures 2 and 3 for Ericsson. Similarly, a time-series plot of the AstraZeneca data is shown in Figure 4. The ACF and PACF plots are presented in Figures 5 and 6, respectively.

**Table 7.** Descriptive statistics for Ericsson and AstraZeneca data.

|  | Mean | Min | Max | Standard Deviation | Skewness | Kurtosis |
|---|---|---|---|---|---|---|
| AstraZeneca | 1.33 | 0.00 | 17.00 | 1.79 | 2.44 | 9.88 |
| Ericsson | 8.12 | 0.00 | 64.00 | 6.36 | 2.05 | 7.16 |

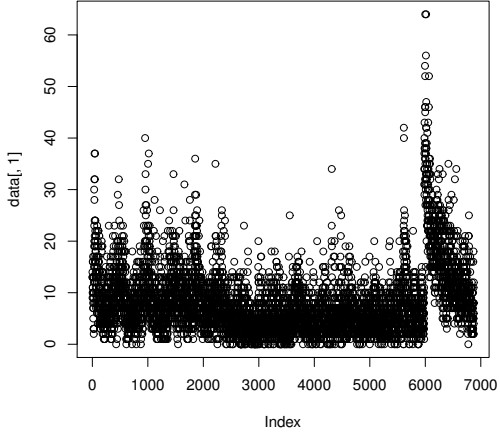

**Figure 1.** Ericsson plot.

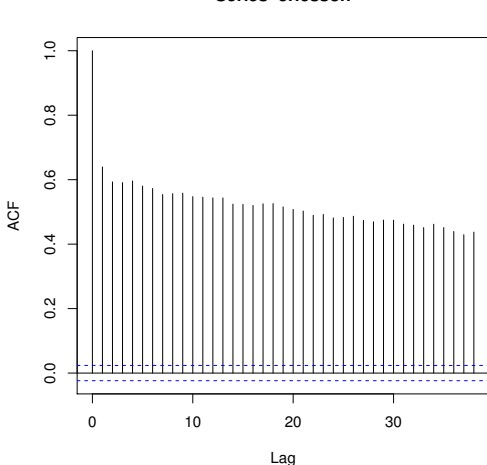

**Figure 2.** ACF for Ericsson plot.

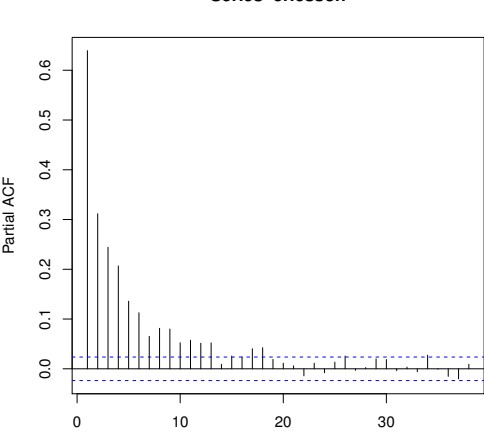

**Figure 3.** PACF for Ericsson plot.

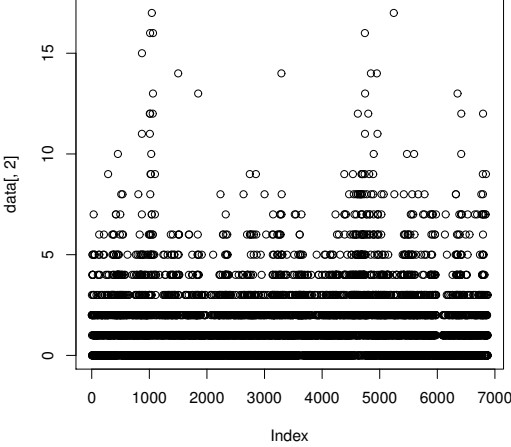

**Figure 4.** AstraZeneca plot.

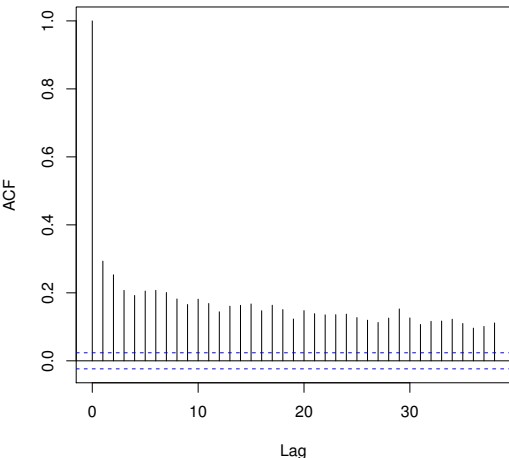

**Figure 5.** ACF for AstraZeneca plot.

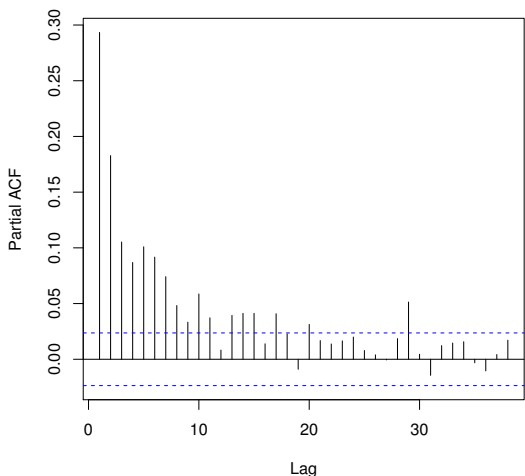

**Figure 6.** PACF for AstraZeneca plot.

From Figures 2 and 5, the Ericsson and AstraZeneca time series both display an almost auto-regressive pattern and their PACFs in Figures 3 and 6 show that the lag-1 autocorrelation has the highest peak, as compared to the other significant lags. However, the sample cross-correlation is nearly $-0.01$.

When applying the BINAR(1) models, we consider the effects of some categorical (0–1) variables such as news $(x_{t,1})$, the Friday effect $(x_{t,2})$, and the time of day effect $(x_{t,3})$, where 0 denotes the reference code. The link predictor can, thus, be expressed as: $\lambda_t^{[k]} = \exp(\hat{\beta}_0^{[k]} + \hat{\beta}_1^{[k]} x_{t,1} + \hat{\beta}_2^{[k]} x_{t,2} + \hat{\beta}_3^{[k]} x_{t,3})$. As seen in Tables 8 and 9 the four bivariate models illustrate that the factors news, Friday effect, and time of day effects are significant and contribute positively to explain the variation in the stock transactions of the two series. We can notice that the BINAR(1)NB and BINAR(1)CMP provide the estimates with lower standard errors than BINAR(1)P and BINAR(1)G. Using the corresponding links in Section 5, the contributory effects can be computed easily. Since BINAR(1)P, BINAR(1)G, and BINAR(1)NB share a common link, it can be noticed that the factors related to Ericsson contribute more to its link predictor than for AstraZeneca. This helps to conclude that investment in Ericsson seems more lucrative.

**Table 8.** Estimates with standard errors under the BINAR(1) models for AstraZeneca (ast) data sets.

| Model | $INTC_{ast}$ | $NEWS_{ast}$ | $FRI_{ast}$ | $TIME_{ast}$ |
|---|---|---|---|---|
| BINAR(1)P | 1.3451 | 0.3141 | 0.2551 | 0.1401 |
| | (0.0431) | (0.0285) | (0.0323) | (0.0444) |
| BINAR(1)G | 1.5612 | 0.3441 | 0.3022 | 0.1221 |
| | (0.0376) | (0.0276) | (0.0229) | (0.0356) |
| BINAR(1)NB | 1.3210 | 0.3562 | 0.2616 | 0.1011 |
| | (0.0253) | (0.0181) | (0.0188) | (0.0152) |
| BINAR(1)CMP | 0.8911 | 0.2991 | 0.3422 | 0.09871 |
| | (0.0389) | (0.0202) | (0.0222) | (0.0112) |

**Table 9.** Estimates with standard errors under the BINAR(1) models for Ericsson (eric) data sets.

| Model | $INTC_{eric}$ | $NEWS_{eric}$ | $FRI_{eric}$ | $TIME_{eric}$ |
|---|---|---|---|---|
| BINAR(1)P | 0.8971 | 0.1021 | 0.3461 | 0.4009 |
| | (0.0431) | (0.0285) | (0.0323) | (0.0444) |
| BINAR(1)G | 0.8612 | 0.1054 | 0.3276 | 0.3201 |
| | (0.0310) | (0.0289) | (0.0277) | (0.0392) |
| BINAR(1)NB | 0.8452 | 0.1101 | 0.3482 | 0.3982 |
| | (0.0312) | (0.0233) | (0.0223) | (0.0310) |
| BINAR(1)CMP | 0.3909 | 0.3561 | 0.4872 | 0.4061 |
| | (0.0502) | (0.0521) | (0.0661) | (0.0321) |

The corresponding over-dispersion indices in Table 10 are proved significant, as well as the thinning dependence coefficients. From Tables 8–10 and Equations (11) and (12), the root mean square errors (RMSEs) and Akaike information criterion (AICs) were computed from the log-likelihood values in the optimum function. The AICS in Table 11 show that BINAR(1)NB and BINAR(1)CMP are more reliable models for the above over-dispersed series. Here, the RMSEs represent the sum of squared differences between true values and one-step conditional expectations using the same training period from 3 to 22 July 2002.

**Table 10.** Estimates and standard errors of the over-dispersion indices and correlations under the BINAR(1) models for AstraZeneca and Ericsson data sets.

| Model | $IOD_{ast}$ | $IOD_{eric}$ | $\rho_{11}$ | $\rho_{12}$ | $\rho_{21}$ | $\rho_{22}$ |
|---|---|---|---|---|---|---|
| BINAR(1)P | | | 0.3501 | 0.1501 | 0.0911 | 0.4771 |
| | | | (0.0887) | (0.0571) | (0.0602) | (0.0687) |
| BINAR(1)G | | | 0.2980 | 0.0909 | 0.1010 | 0.5032 |
| | | | (0.0503) | (0.0455) | (0.0222) | (0.0333) |
| BINAR(1)NB | 0.4322 | 0.1031 | 0.2885 | 0.0888 | 0.0911 | 0.0818 |
| | (0.0853) | (0.0981) | (0.0488) | (0.0252) | (0.0187) | (0.0333) |
| BINAR(1)CMP | 0.7811 | 0.1091 | 0.4422 | 0.09010 | 0.0810 | 0.0761 |
| | (0.1103) | (0.1011) | (0.0321) | (0.0303) | (0.0219) | (0.0551) |

**Table 11.** Goodness-of-fit criteria.

| | **BINAR(1)P** | **BINAR(1)G** | **BINAR(1)NB** | **BINAR(1)CMP** |
|---|---|---|---|---|
| AICs | 2331.210 | 2011.210 | 1890.545 | 1991.321 |
| $RMSE_{ast}$ | 4.321 | 4.051 | 3.761 | 3.788 |
| $RMSE_{eric}$ | 5.011 | 4.899 | 4.320 | 3.891 |

### 6. Conclusions

This paper extends the research findings of Nastic et al. (2016) and illustrates that the family of BINAR(1) models where the respective innovations are independent yield better AICs than the BINAR(1) model, induced by correlated innovations, or those that incorporate both sources of cross-correlation through correlated innovations and previous-lagged survivors. In this work, the proposed BINAR(1)s are adapted to the non-stationary setups with several popular innovation distributions: geometric, negative binomial and COM–Poisson. We also noticed that under mis-specified innovation distributions, the BINAR(1)NB and BINAR(1)CMP yield reliable results. Hence, these two competing models are highly commendable for time series of counts that exhibit simultaneous significant over-dispersion. On the other hand, the paper offers a variety of BINAR(1) models suitable for financial stock series. Last but not least, the proposed models provide the financial analysts insights about the forecast number of transactions subject to the market dynamics.

**Author Contributions:** Conceptualization, Y.S. and N.M.K.; methodology, Y.S.; software, Y.S.; validation, N.M.K.; formal analysis, Y.S.; investigation, Y.S.; resources, N.M.K.; data curation, N.M.K.; writing—original draft preparation, N.M.K.; writing—review and editing, Y.S.; visualization, Y.S.; supervision, N.M.K.; project administration, Y.S.; funding acquisition, N.M.K. All authors have read and agreed to the published version of the manuscript.

**Funding:** The APC was funded by an Author Voucher discount (code:21d367a22f7292b4).

**Data Availability Statement:** Data will be made available on request.

**Conflicts of Interest:** The authors declare no conflicts of interest.

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
