# Peer review of "On Comparing and Assessing Robustness of Some Popular Non-Stationary BINAR(1) Models"

_jrfm, doi:10.3390/jrfm17030100_

Round 1

Reviewer 1 Report

Comments and Suggestions for Authors

The manuscript submitted is well-structured, scientifically clear, and relevant to the journal topic. It presents a set of relevant and up-to-date bibliographical references and the mathematical models that justify the objective being pursued.

However, I believe it relies too heavily on the mathematical model intended to prove robustness in the analysis of the intra-day transaction series in financial companies/companies with financial transactions without properly contextualising the relevance of the mathematical approach in the financial analysis of such transactions. For example, nothing is said in the abstract about the potential conclusions to be drawn from a financial point of view. On the other hand, the introduction starts immediately with the mathematical model without contextualising its importance in the financial context.

This is why I believe the manuscript, which is well presented and descriptive from a mathematical point of view, could be improved by contextualising its relevance from a financial point of view. For this reason, I suggest making minor changes to contextualise the importance of the model and the innovation it demonstrates.

Author Response

Review Report for the Manuscript: Submission ID: jrfm-2888184

On Comparing and Assessing Robustness of Some Popular Non-Stationary BINAR(1) Models

  1. This is why I believe the manuscript, which is well presented and descriptive from a mathematical point of view, could be improved by contextualising its relevance from a financial point of view. For this reason, I suggest making minor changes to contextualise the importance of the model and the innovation it demonstrates.

Response: Necessary amendments have been made in the Abstract (Page 1) in order to contextualise the relevance of the non-stationary BINAR(1) Models in analysing financial series which exhibit significant volatilities and over-dispersion. 

Reviewer 2 Report

Comments and Suggestions for Authors

Dear Author(s),

My evaluation and suggestions about the manuscript are as follows.

Kind regards

·        The paper is well-organized, with a clear structure that guides the reader through the introduction, model construction, parameter estimation, simulation study, and application to real-world data sets.

·         The paper aims to address a gap in the literature regarding the robustness of BINAR(1) models with non-diagonal cross-correlation structures and unpaired innovation series.

·         It presents an approach by focusing on BINAR(1) models with non-diagonal cross-correlation structures and unpaired innovation series. This perspective contributes to the existing literature by offering a new method for analyzing over-dispersion and non-stationarity in time series of counts, distinguishing it from prior works that mostly utilized time-invariant conditions.

·         The study's findings are significant, providing insights into the robustness of BINAR(1) models under different innovation distribution specifications.

·         The methodology is rigorously developed, with a comprehensive simulation study supporting the theoretical findings. The comparison of different BINAR(1) models based on diagnostic measures like RMSE and AICs adds to the paper's scientific validity, providing a robust framework for evaluating model performance.

·         The application of the proposed BINAR(1) models to AstraZeneca and Ericsson data sets demonstrates the practical relevance of the research.

·         It would be beneficial for the authors to further discuss the implications of their findings for future research and practice in the conclusion section to fully capitalize on the study's contributions.

·         There are BINAR(1) processes with paired Poisson, Negative Binomial, Geometric, and COM-Poisson innovations. If there are different probability distribution suggestions to be used for non-stationary BINAR(1), this can be discussed in the conclusion. Discussing alternative distributions in the conclusion could further this discourse. Additionally, elucidating the rationale behind the chosen simulation sizes of 60, 200, and 600 would provide clarity on their significance to the study's robustness.

Author Response

Review Report for the Manuscript: Submission ID: jrfm-2888184

On Comparing and Assessing Robustness of Some Popular Non-Stationary BINAR(1) Models

  1. There are BINAR(1) processes with paired Poisson, Negative Binomial, Geometric, and COM-Poisson innovations. If there are different probability distribution suggestions to be used for non-stationary BINAR(1), this can be discussed in the conclusion. Discussing alternative distributions in the conclusion could further this discourse. Additionally, elucidating the rationale behind the chosen simulation sizes of 60, 200, and 600 would provide clarity on their significance to the study's robustness.

Response: Necessary amendments have been made in the Conclusion (Page 9) in order to further discuss the implications of the findings for future research on the suitability of BINAR(1) models for financial stock series and to highlight the financial analysts insights to the market dynamics.
